# Evaluating the Phenotypic and Genomic Characterization of Some Egyptian Phages Infecting Shiga Toxin-Producing *Escherichia coli* O157:H7 for the Prospective Application in Food Bio-Preservation

**DOI:** 10.3390/biology11081180

**Published:** 2022-08-05

**Authors:** Dina El-Sayed, Tarek Elsayed, Nadia Amin, Ahmad Al-Shahaby, Hanan Goda

**Affiliations:** Department of Agricultural Microbiology, Faculty of Agriculture, Cairo University, Giza 12613, Egypt

**Keywords:** Shiga toxin-producing *Escherichia coli*, bacteriophage, food bio-preservation, biological characterization, genome sequence

## Abstract

**Simple Summary:**

Shiga toxin-producing *Escherichia coli* (STEC) represents a hazardous health problem because it causes various human gastrointestinal tract diseases, for example, bloody diarrhea and hemorrhagic colitis. The major concern of STEC O157:H7 resulted from its biological characteristics, including low infective dose, ability to express different virulence factors and multidrug resistance of some species. Principally, the human outbreaks of STEC O157:H7 are associated with consumption of undercooked or contaminated bovine dairy and meat products. Treatments of *E. coli* infections have been increasingly complicated as a result of the development of antibiotic resistance. For this reason, as well as the increasing consumer demand for safe food products, it has become important to apply alternative effective and eco-friendly approaches, such as using lytic phages, to control the growth of pathogenic bacteria in food. This study focused on evaluating the applicability of locally isolated lytic phages specific to Shiga toxin-producing *Escherichia coli* O157:H7 as prospective biocontrol agents in food. Our findings presented two phages with promising biological and genomic characteristics to be applied in food bio-preservation.

**Abstract:**

Shiga toxin-producing *E. coli* (STEC) is considered a worldwide public health and food safety problem. Despite the implementation of various different approaches to control food safety, outbreaks persist. The aim of study is to evaluate the applicability of phages, isolated against STEC O157:H7, as prospective food bio-preservatives. Considering the relatively wide host range and greatest protein diversity, two phages (STEC P2 and P4) from four were furtherly characterized. Complete genome analysis confirmed the absence of toxins and virulence factors—encoding genes. The results confirmed the close relation of STEC P2 to phages of *Myoviridae*, and STEC P4 to the *Podoviridae* family. The phages retained higher lytic competence of 90.4 and 92.68% for STEC P2 and P4, respectively with the HTST pasteurization. The strong acidic (pH 1) and alkaline (pH 13) conditions had influential effect on the surviving counts of the two phages. The lowest survivability of 63.37 and 86.36% in STEC P2 and P4 lysate, respectively appeared in 2% bile salt solution after 3 h. The results confirmed the strong effect of simulated gastric fluid (SGF) on the survivability of the two phages comparing with simulated intestinal fluid (SIF). Therefore, the two phages could be applied as a natural alternative for food preservation.

## 1. Introduction

Generally, *Escherichia coli* (*E. coli*) is an inoffensive commensal of the healthy intestinal tract of humans and animals. However, some types of *E. coli* are pathogens with the ability to cause food-borne diseases. There are six pathovars of disease-causing *E. coli* from which enterotoxigenic *E. coli* (ETEC), enterohaemorrhagic *E. coli* (EHEC) or verocytotoxin-producing *E. coli* (VTEC) that cause diarrheal illness [1].

Shiga toxin-producing *E. coli* (STEC) are classified as enterohaemorrhagic *E. coli*, and represent a dangerous public health problem worldwide because it causes various human gastrointestinal tract diseases, comprising bloody diarrhea and life-threatening diseases, such as thrombotic thrombocytopenic purpura, (TTP), hemorrhagic colitis and hemolytic-uremic syndrome (HUS) [2]. The most important STEC serotypes associated with food poisoning outbreaks in the world are O157, O146, O145, O128, O121, O118, O117, O113, O111, O103, O91 and O26 [3].

The major concern of STEC O157:H7 worldwide resulted from its biological characteristics including low infective dose, ability to express different virulence factors, multidrug resistance of some species and long survival time in the environment [4,5]. This microbe is naturally widespread in beef and dairy cattle. Therefore, the human outbreaks are associated with the consumption of undercooked or contaminated bovine dairy and meat products. People can also be infected with STEC through drinking contaminated water, consuming contaminated raw vegetables, or through contact with animal feces [6].

The different physical and chemical methods applied to control and reduce the risk of pathogenic bacteria, such as STEC, in food have not been sufficiently efficient, since outbreaks continue to happen. In addition, some methods may change the organoleptic characteristics of food. The problem is made increasingly sophisticated because of the resistance of pathogenic bacteria to the antibiotics. Treatments of *E. coli* infections have been increasingly complicated as a result of the development of antibiotic resistance, especially to β-lactams (as penicillins, cephalosporins, cephamycins, and carbapenems), tetracycline and aminoglycosides. The excessive and inappropriate use of antibiotics to treat animal infections promoted resistance in commensal and pathogenic bacteria. It has been suggested that these bacteria will be a reservoir of resistance genes for other bacteria, leading to the prevalence of multidrug resistant bacteria [7]. Due to the above reported facts, and the increasing consumer demand for food products free of synthetic chemicals, it has become essential to apply alternative approaches to control the growth of pathogenic bacteria. The use of bacteriophages is recommended as a safe, effective and environmentally friendly strategy in food safety [8].

The lytic bacteriophages have many significant traits making them a promising antibacterial agent to be applied against bacterial pathogens in food. They display a very limited host range even among specific bacteria and bacterial strains, meaning they do not disrupt the normal microflora in humans. They do not infect or colonize the eukaryotic cells and do not induce harmful effects in human health [9]. The phages do not have undesirable effects on chemical or physical food quality because they do not produce any substances to change the food characteristics [10]. Furthermore, phages are ubiquitous in different ecosystems, so it is relatively easy to isolate them from the environment [11].

Some studies have shown high efficiency in the use of phages against *E. coli* O157:H7 in foods such as tomatoes, broccoli [12], beef [13], cantaloupe [14], and leafy green vegetables [15].

The Food and Drug Administration (FDA) regulated the use of phage preparations as a food additive, for the first time, on 18 August 2006. On this date, the FDA declared its approval of the use of a bacteriophage preparation on ready-to-eat (RTE) meat and poultry products against *Listeria monocytogenes* [16]. On 8 August 2017, the FDA described the phage preparation GRN 724 as GRAS and approved its use as a processing aid antimicrobial agent against Shiga toxin-producing *E. coli* on the surface of beef carcasses [17].

The intended objective of the current study is to isolate locally lytic phages specific to Shiga toxin-producing *E. coli* O157:H7, and to study their applicability as prospective biocontrol agents in food preservation through an evaluation of their biological characterization and complete genome sequence.

## 2. Materials and Methods

### 2.1. Bacterial Strain

The Shiga toxin-producing *Escherichia coli* O157:H7 (STEC O157:H7) wild type strain 93,111, obtained from National Research Centre (NRC), Giza, Cairo, Egypt, was used as a representative enterohaemorrhagic *E. coli* for phage isolation. 

Initially, the presence and expression of major virulence genes of STEC; Shiga toxin (*stx*) encoding-genes (*stx1* and *stx2*) and intimin-encoding gene (*eae*) were confirmed in the selected strain. These genes were detected in DNA extracted from the pure culture applying the standard method that uses real-time PCR as a reference technology for detection of virulence and serogroup-associated genes [18]. The results were confirmed by measuring the expression of these genes as follows: *E. coli* cells were grown in tryptone soy broth (TSB) for 18 h at 37 °C with shaking at 180 rpm. Subsequently, 40 μL broth culture were introduced into 4 mL TSB and incubated for 4 h at 37 °C with shaking at 180 rpm [19]. After that, the bacterial pellet was obtained by centrifugation at 5000× *g*/5 min. According to the manufacturer’s instructions, total RNA was isolated using a bacterial RNA extraction Kit (GeneJET RNA 110 Purification Kit, Thermo Fisher Scientific, Bremen, Germany) and purified using Dnase I, Rnase-free kit (Thermo Scientific, Bremen, Germany) to remove genomic DNA from extracted RNA. The purity and concentration of RNA were measured using UV-Vis NanoDrop spectrophotometer (NanoDrop 2000, Thermo Fisher Scientific, Bremen, Germany). The integrity of RNA was checked by running 5 μL on 1% agarose gel electrophoresis in 0.5 × TBE-buffer for 1 h (50 V). The extracted RNA samples were stored at −70 °C until use. The conversion of RNA to complementary DNA was performed using RevertAid First Strand cDNA Synthesis (Thermo Fisher Scientific). To detect *stx1*, *stx2* and *eae* transcript levels in STEC O157:H7 wild type strain 93,111 and *E. coli* ATCC 35218, real-time quantitative PCR (qPCR) was performed. The real-time PCR was carried out on a StepOnePlus Real-Time PCR System (Applied Biosystem, S/N 272008693). The amplification reactions were achieved in 20 μL containing 4 μL of 5× HOT FIREPol^®^ EvaGreen^®^ qPCR Mix Plus (ROX) (Solis BioDyne), 0.5 μL of each forward and reverse primer and 1 μL of cDNA template. The volume was completed up to 20 μL using nuclease-free water to adjust the concentration of primers at 2 mM. Thermal cycling conditions included initial denaturation temperature of 95 °C for 12 min, followed by 40 cycles of amplification for15 s at 95 °C, annealing for 30 s at 60 °C and extension for 30 s at 72 °C for each cycle. The 40 cycles were followed by melt curve analysis which involved heating to 95 °C for 1 min, followed by cooling to 55 °C for 30 s and heating to 95 °C while monitoring fluorescence. 16S rRNA gene was used as a housekeeping gene. The gene-specific primers used to detect the expression of selected virulence genes were *stx1*-F (ATAAATCGCCATTCGTTGACTAC) and *stx1*-R (AGAACGCCCACTGAGATCATC) for *stx1* gene, *stx2*-F (GGCACTGTCTGAAACTGCTCC) and *stx2*-R (TCGCCAGTTATCTGACATTCTG) for *stx2* group gene, *eae*A-F (GACCCGGCACAAGCATAAGC) and *eae*A-R (CCACCTGCAGCAACAAGAGG) for *eae* gene. The primers of 16S rRNA gene were Eub338 (ACTCCTACGGGAGGCAGCAG) and Eub518 (ATTACCGCGGCTGCTGG).

This strain (STEC O157:H7 wild type strain 93,111) was also tested for the lysogeny through irradiation of exponentially growing cells (10^8^ cfu/mL TSB) with UV light with wavelength of 254 nm/30 s and 1 min. The irradiated cells were cultivated using tryptone soy agar (TSA) with incubation at 37 °C/24 h. Appearance of the bacterial lawn indicates absence of prophage in the tested *E. coli* strain.

Finally, the non-lysogenic STEC O157:H7 was maintained on TSA at 4 °C and transferred periodically for further studies.

### 2.2. Water Samples Collection and Preparation

Different water samples were screened for the presence of phages infecting Shiga toxin-producing *E. coli* O157:H7. Fifteen water samples (500 mL each) comprised 11 sewage, and 4 tap water samples were collected from Giza Governorate using sterile 500 mL bottles. Prior to phage isolation, the sewage samples were filtered through filter paper to remove the impurities.

### 2.3. Phage Isolation and Purification

Generally, the phage enrichment is a very important step in facilitating phage isolation. Ten mL of tap water or filtered sewage were added to 50 mL TSB inoculated with 5 mL of log phase STEC O157:H7 broth culture and incubated at 37 °C/24 h with shaking at 100 rpm. After the enrichment, the supernatant was collected by centrifugation at 6000 rpm/15 min, followed by filter sterilization through membrane filter with 0.45 µm pore size to remove any bacterial cell debris. The filtrate was assessed for the presence of phages applying the spot test in which five µL of filtrate were spotted on TSA layer inoculated with STEC O157:H7. Appearance of plaques on the bacterial lawn was used as an indicator for the presence of targeted lytic phages. The plaques observed on TSA layer were picked up, suspended in STEC O157:H7 broth culture, incubated at 37 °C/24 h with shaking at 100 rpm to release the phage particles from agar and centrifuged at 6000 rpm/15 min. The obtained supernatant was filtrated through 0.45 µm pore size membrane filter and purified applying the double agar layer technique [20].

In this technique, serial ten-fold dilutions of the phage supernatant were prepared in 1× phosphate buffer (pH 7.2). Definite volume from each dilution (0.5 mL) was mixed with 0.5 mL freshly prepared STEC O157:H7 broth culture (10^7^–10^8^ cfu/mL). This mixture was added to 3.0 mL of melted semi-solid TSA (0.7% agar) and poured on pre-solidified TSA base layer (2.0% agar). The plates were incubated at 37°C/24 h. The pure phage suspension (phage lysate) was prepared from single plaque as mentioned before and stored at 4 °C. 

To perform all characterization experiments, the pure, fresh, and high titer phage lysate (10^9^–10^11^ pfu/mL) was prepared, and the phage titer (pfu/mL) was determined by double agar layer technique. 

### 2.4. Phenotypic Phage Characterization

#### 2.4.1. Morphological Characterization

All purified phages were negatively stained by 2% (*w*/*v*) phosphotungstate at pH 7.2. The stained phage particles were examined microscopically by transmission electron microscope JEOL (JEM-1400 TEM). Images were captured by CCD camera model AMT, Optronics camera with 1632 × 1632 pixel format as side mount configuration. 

#### 2.4.2. Determination of Structural Proteins 

Fifteen milliliters of each phage were subjected to a concentration ultracentrifugation process using sorvall MTX 150 mini-ultracentrifuge (Thermo Fisher Scientific) (100,000× *g*, 4 °C, 2 h) in 20% sucrose (cushion method). Phage pellets were resuspended in 0.5 mL PBS, and the total protein concentration was assessed using Pierce™ BCA Protein Assay (Thermo Fisher Scientific Kit). Proper volume of each phage suspension with a concentration of 200 μg/mL was mixed with 4× “Laemmli buffer” with 10% beta-mercaptoethanol, then subjected to 95 °C for 5 min, cooled on ice for 1 min and centrifuged shortly. PageRuler Prestained Protein Ladder (Thermo Fisher Scientific) was loaded as a marker. Twenty microliters of each phage suspension were resolved on precast gradient NuPAGE Bis-Tris gel system 4–12% (Invitrogen, Darmstadt, Germany) using X Cell Sureiock Mini-Cell electrophoreses system (Invitrogen) (150 V current for 2 h). The resolved SDS-page was fixed for 30 min (ethanol 30% acetic acid 10%), then conditioned in 20% ethanol for 30 min and rained with water for additional 4 times. Coomassie Brilliant Blue staining procedure was used for protein visualization. Briefly, staining 20 min step (0.1% *w*/*v* Coomassie blue, 20% *v*/*v* methanol and 10% *v*/*v* acetic acid), followed by distaining step (50% *v*/*v* methanol and 10% acetic acid), then stopped in 5% *v*/*v* acetic acid storage solution. Stained gel was imaged using the iBright FL1500 Imaging System (thermo-Fisher Scientific), and the captured images were analyzed using the iBright web-based app (https://apps.bisher.com/apps/spa/) (accessed on 2 September 2021).

#### 2.4.3. Host Range Analysis

The host range of all purified phages was determined applying the spot test against 25 Gram-negative bacterial strains. These strains were categorized into three classes: Shiga toxin-producing *Escherichia coli* (7 strains), nontoxigenic *Escherichia coli* (6 strains) and other G^-^ bacteria from Enterobacteriaceae (12 strains including 1 *Salmonella typhimurium*, 1 *Shigella sonni*, 1 *Shigella boydii,* 2 *Klebsiella pneumoniae,* 1 *Klebsiella quasipneumoniae,* 1 *Enterobacter aerogenes,* 1 *Enterobacter ludwigii,* 1 *Enterobacter asburiae,* 1 *Enterobacter hormaechei* subsp. Xiangfangensis and 2 *Enterobacter cloacae*). The plates were checked for plaques formation after incubation at 37 °C/24 h. The presence of a lytic zone on the lawn of tested bacteria was considered as an evidence of phage susceptibility.

The phages with relatively wide host range and greatest protein diversity were characterized furtherly through determination of lytic activity and multiplicity of infection (MOI). These phages were also evaluated as a potential biocontrol agent to be used in food bio-preservation. Additionally, the complete genome sequence of selected phages was analyzed. All experiments were conducted in triplicates.

#### 2.4.4. Determination of Multiplicity of Infection (MOI)

The overnight broth culture of STEC O157:H7 was diluted to a count of 1.5 × 10^8^ cfu/mL using TSB. The bacterial cells were infected with each selected phage individually at different ratios represented by 1, 0.1, 0.01 and 0.001 pfu/cfu. After incubation at 37 °C/4 h with shaking at 100 rpm, the mixture was centrifuged at 6000 rpm/15 min, and the supernatant was filtrated through 0.45 µm pore size membrane filter to determine the phage titer [21].

#### 2.4.5. Determination of Lytic Activity

The lytic activity of selected phages against STEC O157:H7 wild type strain 93,111 was evaluated with tested MOIs. During incubation at 37 °C, the count of *E. coli* was determined at 1 h intervals for 7 h applying the pour plate method using TSA [22].

#### 2.4.6. UV Radiation Stability

The phage UV stability was assessed by exposure of phage lysate (5 mL in open plate) to the UV light with a wavelength of 254 nm/15, 30, 45, 60 and 75 min. The survival of tested phages was determined by double-layer agar technique. 

#### 2.4.7. Thermal Stability

The thermal stability of phage particles was evaluated applying different heat treatments employed in food processing. The phage lysate was heated at 100 °C/10, 20 and 30 min, 63 °C/30 min (low temperature long time pasteurization, LTLT) and 72 °C/15 s (high temperature short time pasteurization, HTST). The phage survival was determined by double-layer agar technique.

#### 2.4.8. pH Stability

The pH survivability of phage was investigated through introducing the phage suspension to different pH values e.g., 1, 3, 5, 7, 9, 11 and 13 with incubation at 37 °C for 1 h. The TSB was pre-adjusted to tested pH values by 1 M HCl or NaOH, and then 1 mL of the phage lysate was inoculated into 9 mL of TSB. The titer of surviving phages was measured by the overlay technique.

#### 2.4.9. Bile Salt Stability

The phage stability in 1 and 2% (*w*/*v*) bile salt was assessed through adding 1 mL of phage lysate (10^10^ pfu/mL) to 9 mL of bile salt solution. After incubation at 37 °C, the phage titer was determined after 1 and 3 h. Sterile distilled water at pH 7 was used as a control [8].

#### 2.4.10. Simulated Gastric Fluid (SGF) Stability

The gastric digestion of phage particles was evaluated by adding 1 mL of phage lysate to 9 mL of SGF prepared by dissolving 0.13 g NaCl, 0.64 g Na HCO_3_, 0.024 g KCl and 0.3 g bile salt in 50 mL sterile distilled water. After pH adjustment to 2.5, 1 g pepsin (Loba Chemie PVT. LTD., Mumbai, India) was dissolved, and the total volume was completed up to 100 mL [23]. The mixture was incubated at 37°C, and the phage titer was estimated after 15, 30, 60, 120, 180 and 240 min. Distilled water was used as a control. 

#### 2.4.11. Simulated Intestinal Fluid (SIF) Stability

The phage stability in simulated intestinal fluid (SIF) was estimated through inoculating 1 mL of phage lysate in 9 mL of (SIF) formulated by dissolving 0.68 g KH_2_PO_4_ in 25 mL sterile distilled water. After adjusting the pH to 6.8, 1 g pancreatin (Loba Chemie PVT. Ltd., India) was dissolved, and the total volume was completed to 100 mL [24]. The phage titer was determined after incubation at 37 °C/15, 30, 60, 120, 180 and 240 min.

### 2.5. Genome Sequencing and Analysis 

A total of 200 μL of sucrose cushion concentrated phage were subjected to nucleic acid extraction using GeneJET™ Viral DNA and RNA Purification Kit (Thermo Fisher Scientific) according to the manufacturer instructions. The phage DNA concentration was measured using Qubit 3 Fluorometer (Thermo Fisher Scientific, Waltham, MA, USA) using Qubit^TM^ dsDNA HS Assay Kit according to manufacturer instructions. Libraries preparation of phage DNA was conducted using Ion Xpress^TM^ Plus Fragment Library Kit (Thermo Fisher Scientific, Waltham, MA, USA). Then, the plate was prepared using Ion 520^TM^ Kit-OT2. Finlay, the libraries were loaded on the Chip of ion S5^TM^ System. Reads were trimmed and assembled into a whole genome using the PATRIC bioinformatics resource center online platform (https://www.patricbrc.org) (accessed on 15 October 2021) after checking reads quality using the FastQC software version 0.11.4 (https://www.bioinformatics.babraham.ac.uk/projects/fastqc) (accessed on 15 October 2021) (Babraham, Cambridge, UK). 

### 2.6. Phage Genome Annotation 

The phages were annotated using RAST tool kit (RASTtk). Functional annotation was performed using the BLAST tools (BLASTn and BLASTp) at NCBI (http://blast.ncbi.nlm.nih.gov/Blast.cgi) (accessed on 20 October 2021) as well as the BLAST search tool of KEGG (BlastKOALA; http://www.kegg.jp/blastkoala) (accessed on 20 October 2021) [25] to identify each gene through the genome. Transfer RNAs were identified using tRNA scan-SE (http://lowelab.ucsc.edu/tRNAscan-SE/) (accessed on 20 October 2021). 

### 2.7. Comparative Genomics 

The whole nucleotide sequence similarities between phages were determined by megablast analysis at NCBI. Comparison of open reading frames (ORFs) from relative phages was performed using EasyFig software version 2.2.5 (https://mjsull.github.io/Easyfig/files.html) (accessed on 25 October 2021). Pairwise comparisons of the nucleotide sequences were conducted using the Genome BLAST Distance Phylogeny (GBDP) method [26] using recommended settings for prokaryotic viruses [27]. To infer a balanced minimum evolution tree, the resulting intergenomic distances were used. In silico DNA-DNA hybridization (isDDH), values were calculated by the Genome-to-Genome Distance Calculator (GGDC 2.1) (http://ggdc.dsmz.de/distcalc2.php) (accessed on 25 October 2021) using the recommended BLAST+ alignment and formula2 (identities/HSP length) [26]. Average nucleotide identity (ANI) between tested phages and the closest phages, obtained from the NCBI, was calculated using the JSpeciesWS Online Service (http://jspecies.ribohost.com/jspeciesws/#analyse) (accessed on 5 November 2021) [28]. Whole genome comparisons were performed using the BRIG software v. 0.95 (BLAST Ring Image Generator) (http://sourceforge.net/projects/brig) (accessed on 5 November 2021) using *Escherichia* phage vB-EcoM-PhAPEC2 representing the closest phage to ours as a reference genome [29].

### 2.8. Phylogenetic Analysis

Phylogenetic analyses between the genome of related phages were performed using MEGA software 5.0 using the Neighbor-Joining Algorithm. Closely related genomes were identified applying BLAST search. The *terL* gene, encoding the large subunit of terminase, was used to construct phylogenetic tree comparing tested phages with all the closely related phages obtained from the NCBI database. This was followed by phylogenetic analysis based on the whole genome sequence of the most closely related phage genomes. 

The whole genome sequence of two selected phages (STEC P2 and STEC P4) was deposited in NCBI GeneBank with the name of *Escherichia* phage ST2 and *Escherichia* phage ST4, respectively, under accession number of OM982647 and OM982646, respectively. 

### 2.9. Statistical Analysis

A randomized complete block design with two factors was used for analysis of all data with three replications for each parameter. The treatment means were compared by least significant difference (L.S.D.) test using Assistat program [30].

## 3. Results

The selected STEC (Shiga toxin-producing *E. coli* O157:H7 wild type strain 93,111) was an identical host to isolate the target lytic phages. This was confirmed through evaluating the presence and expression of the main virulence genes contained within it, and testing its lysogeny state. The RT-PCR results confirmed the presence and successful expression of *eae*, *stx1* and *stx2* genes (Appendix A). The results also confirmed that the bacterial host was non lysogenic (free of prophage) because it did not exhibit plaques on the bacterial lawn after UV irradiation. 

### 3.1. Isolation of Enterohaemorrhagic E. coli O 157:H7 Infecting Phages

The screening of water samples yielded a total of eleven crude lysates active against Shiga toxin-producing *E. coli* O157:H7 wild type strain 93,111, as the phages were recovered from all tested sewage samples.

Based on formation of a single clear plaque on the host bacterial lawn in the purification process, and the possibility of preparing a high titer of the phages, four phages were selected and designated as STEC P1-STEC P4 for further characterization. These phages were furtherly purified and propagated to obtain a high titer of 10^9^–10^11^ pfu/mL. 

### 3.2. Phage Morphology

The TEM analysis of phage virions in Figure 1 revealed that STEC P1 possesses tailless icosahedral capsid with pentagonal outlines of dimensions 137 × 144 nm. The hexagonal head with dimensions of 93 × 113 nm and long tail with a length of 123 nm were specified for STEC P2. The morphology of STEC P3 was characterized as tailless 6—icosahedral capsid with dimensions of 81 × 70 nm. The STEC P4 has a 6—icosahedral head with dimensions of 87 × 86 nm, and a short tail with a length of 30 nm.

### 3.3. Phage Structural Proteins 

SDS-PAGE was applied to analyze the structural proteins profile corresponding to the selected phages (Figure 2 and Appendix A). The virions of STEC P1 and P3 have 13 structural proteins with a molecular weight ranged from 1.32–119.22 and from 9.44–122.39 kDa, respectively. The largest number of structural proteins was recorded in STEC P2 (14 proteins with a molecular mass of 1.32–120.63 KDa) and STEC P4 (17 proteins with a molecular mass of 6.62–142.51 Kda).

### 3.4. Host Range Assay

Spot test performed to evaluate the host range of the four selected phages revealed that all phages can infect different Shiga toxin-producing *E. coli* including STEC O157:H7 ATCC 35150, O157 86.24 and O103 87-293. STEC P2 and P4 also have the ability for lysis of STEC O26 Decaf and O145 6940, respectively (Table 1). Furthermore, for nontoxigenic *E. coil*, the phages have a lytic activity against 16.67% for STEC P1 and P4, 50 and 33.33% for STEC P2 and P3, respectively. Interestingly, only STEC P2 displayed a lytic activity against *Shigella boydii* (*S. boydii*). No phages exhibited activity against tested strains of *Salmonella*, *Klebsiella* and *Enterococcus*. 

According to the results of the host range determination and structural proteins analysis, STEC P2 and P4 were selected for further studies as they have the highest ability for lysis of different STEC (57.14%) and greatest protein variation. Additionally, STEC P2 has a lytic activity against another food borne pathogen *Shigella boydii*.

### 3.5. Determination of MOI and Lytic Activity

Multiplicity of infection (MOI) is defined as the ratio of phage particles to bacterial host cells. In this study, a definite volume of the STEC O157:H7 wild type strain 93,111 culture containing 1.5 × 10^8^ cfu/mL was infected with different volumes of STEC P2 and P4, individually, to obtain a ratio of 1, 0.1, 0.01 and 0.001 pfu/cfu (Table 2). 

After 4 h of incubation at 37 °C, it was observed that there was a significant difference between all MOI groups of the two phages, except among groups 2 and 3 for the STEC P2. The MOI of 1 and 0.1 yielded the highest average phage progeny of 3.5 × 10^9^ and 4.3 × 10^10^ pfu/mL, respectively, for STEC P2, and 4.7 × 10^10^ and 8.7 × 10^9^ pfu/mL, correspondingly for STEC P4.

The lytic activity of STEC P2 (Figure 3) and P4 (Figure 4) against their normal host was estimated at the evaluated MOIs for 7 h.

At each time interval, decreasing the count of STEC cells in infected cultures, with different MOIs, was observed comparing with the phage—free bacterial culture. After one hour, there was a significant decrease in the STEC cells count in phage infected cultures with MOIs of 0.1 and 1 pfu/cfu, and a sharp decrease was observed with MOI of 1. With MOI of 1, the average counts reduced from 8.27–6.7 and from 7.2–4.26 log cfu/mL in the cultures infected with STEC P2 and P4, respectively. The selected phages displayed a complete lysis efficiency of bacteria after 6 h of infection with STEC P2 and P4 with MOI of 0.1 and 1, respectively. STEC P4 with MOI of 0.1 also had the ability to lyse bacterial cells comprehensively after 7 h. Conversely, at the same tested time, the average count of *E. coli* cells reduced by 96.01% in the presence of STEC P2 with MOI of 1.

### 3.6. Effect of UV Radiation on Phage Stability

The effect of UV radiation was evaluated by exposing STEC P2 and P4 virions to UV light (254 nm) for 75 min (Table 3). The results confirmed that the phage survival is inversely proportional with exposure time to UV. For individual phages, there were no significant differences in virions count after exposure to UV radiation for 60–75 min. Conversely, comparing plaque counts of the two irradiated phages revealed that there was a significant difference in the count of surviving virions after 60 and 75 min exposure to UV. The results suggested that the STEC P4 was more resistant to UV light as the highest reduction percentage of virions count was estimated for STEC P2. After 75 min, 89.11 and 71.51% of the treated STEC P2 and P4, respectively, were inactivated. 

### 3.7. Thermal Stability

To evaluate the phage stability under high temperatures, the virions of STEC P2 and P4 were heated, applying HTST, LTLT pasteurization and boiling for 10, 20 and 30 min (Table 4). The results indicated that there was a significant decrease in the virions count of STEC P2 after pasteurization and boiling. Additionally, there was no significant difference between the counts of surviving virions of STEC P4 after pasteurization. It was observed that the LTLT pasteurization had a strong killing effect comparing with HTST method as the greatest reduction percentage of STEC P2 (53.1%) and STEC P4 (18.89%) was estimated after LTLT method. The tested phages lost their lytic activity entirely after boiling for 10, 20 and 30 min.

### 3.8. pH Stability

The phage stability under acidic and alkaline conditions was quantitatively investigated (Table 5). The results showed significant differences in lytic capacity between STEC P2 and P4 after subjecting to pH conditions of 1, 3, 5, 9, 11 and 13 for 1 h at the ambient temperature. It was observed that the strong acidic (pH 1) and alkaline (pH 13) conditions had an influential effect on the surviving counts of phages. STEC P2 lost the ability to infect its host completely at pH 1 and 80.62% at pH 13. Conversely, there was no lytic activity observed from STEC P4 at pH 13. 

### 3.9. Phage Stability in Bile Salt

Survivability of the two phages in 1 and 2% bile salt solution was assessed after 1 and 3 h (Figure 5 and Figure 6). Statistical analysis showed no significant difference in the survivability of each individual phage in 1% bile salt for 1 or 3 h, whereas there was a significant difference in time with a concentration of 2%. There was also no significant difference between the two concentrations for 1 h in the survivability of STEC P2 only. The lowest survivability of 63.37 and 86.36% in STEC P2 and P4 lysate, respectively, appeared in 2% bile salt solution after 3 h. Conversely, the greatest survivability was recorded in 1% bile salt solution after 1 h (94.88% for STEC P2 and 96.83% for STEC P4). 

### 3.10. Phage Stability in Simulated Gastric (SGF) and Intestinal Fluids (SIF)

The stability of STEC P2 and P4 virions in simulated gastric and intestinal fluids was assessed in time intervals for 4 h.

In SGF (Figure 7), statistical analysis confirmed the presence of significant difference in each time between the two phages for 2 h. After 15 min, the count of surviving virions of STEC P2 and P4 was 6.77 ± 0.11 and 7.21 ± 0.35 log pfu/mL with average survival rates of 94.55 and 92.2%, respectively. After 4 h, the survivability decreased to 80.17 and 73.41% for STEC P2 and P4, respectively. 

In SIF (Figure 8), a significant difference was shown between the two phages after 30, 120 and 180 min only. For 15 min, STEC P2 retained its viability completely, and started to reduce after 30 min. The average survival rate decreased to the lowest level of 87.53% after 3 h and 87.45% after 4 h in STEC P2 and P4, correspondingly.

Finally, the results confirmed the strong effect of SGF against the survivability of two phages comparing with SIF. 

### 3.11. Genome Characterization

To realize the phage–host interaction, and to confirm the absence of both toxins and virulence factors, encoding genes for food application, the whole genome of STEC P2 (*Escherichia* phage ST2) and P4 (*Escherichia* phage ST4) was completely analyzed. The whole genome was assembled using the PATRIC bioinformatics platform, and annotated using RAST tool kit (RASTtk). 

*Escherichia* phage ST2 genome was assembled into four contigs with a size of 171,086 bp and an average GC content of 37.97%. This genome has 270 protein coding sequences (CDS), three transfer RNA (tRNA) genes and zero ribosomal RNA (rRNA) genes. Genome annotation revealed that, among the CDS, 41 (15.19%) and 229 (84.81%) are expressed to produce hypothetical and functional proteins, respectively. The proteins with functional assignments included 16 proteins with enzyme commission (EC) numbers, 16 with gene ontology (GO) assignments and 10 proteins that were mapped to KEGG pathways. Alternatively, *Escherichia* phage ST4 genome was assembled into one contig with a size of 72,213 bp and an average GC content of 42.99%. This genome has only 85 protein coding sequences (CDS), two transfer RNA (tRNA) genes and zero ribosomal RNA (rRNA) genes. The CDS included 55 (64.71%) and 30 sequences (35.29%) predicted to produce hypothetical and functional proteins, respectively. The functional proteins were four proteins with enzyme commission (EC) numbers, four with gene ontology (GO) assignments, and four proteins that were mapped to KEGG pathways. A circular graphical display of the genome annotations distribution is provided in Figure 9. This includes genes encoding structural proteins (phage head, tail, neck and base plate), hypothetical proteins, phage proteins with unknown functions and genes coding for enzymatic activities and regulation mechanisms.

The functional annotation was performed using both BLASTn and BLASTp, at the NCBI, as well as BLAST search tool of KEGG to identify each gene through the genome and to study the presence or absence of genes or gene clusters associated with virulence and toxins as well as insertion sequences. The results of genome analysis based on BLAST confirmed that both phages do not encode known genes associated with toxins or other virulence factors. Furthermore, no insertion sequences were detected in the two phage genomes. 

### 3.12. Comparative Genome Analysis

Similarities of the whole viral nucleotide sequence between phages were determined using megablast analysis at NCBI. *Escherichia* phage ST2 showed high similarity (≥85% coverage and 95.4–98.92% identity) to 44 phages isolated from different regions around the world. This phage was closely related to *Escherichia* phage vB-EcoM-PhAPEC2 (98.92% similarity and 96% query cover) (Appendix A) isolated from river water in Belgium and had lytic activity against 19 out of 31 tested Avian pathogenic *Escherichia coli* including serotypes (O45, O78, O2, O18 and O83) [31]. Furthermore, this phage exhibited high similarity with phages infecting other genera related to *Escherichia*, such as *Shigella*. The identity of 97.38, 96.44 and 96.33% were observed between *Escherichia* phage ST2 and each of *Shigella* phage Shf125875 (accession number KM407600.1), *Shigella* phage phi25-307 (accession number MG589383.1) and *Shigella* phage JK42 (accession number MK962756.1), respectively (Appendix A). *Escherichia* phage ST2 also showed 97.47% similarity to Enterobacteria phage RB69 (accession number AY303349.1).

*Escherichia* phage ST4 showed similarity ranging from 92 to 95% with an average coverage between 81 and 87% to 26 phages. The closest hits were *Escherichia* phage vB EcoP PhAPEC7, vB EcoP PhAPEC5 and vB EcoP G7C with 95% similarity and accession number of KF562340.1, KF192075.1 and HQ259105.1, respectively (Appendix A). This phage also displayed an identity of 93% with Enterobacteria phage Bp4 (accession number KJ135004.2) and ECBP1 (accession number JX415535.1). For genomic comparison, the nine closest phages to the two tested phages (ST2 and ST4) from the NCBI GenBank were selected (Appendix A). The obtained ANI (average nucleotide identity) and is DDH (in silico DNA–DNA hybridization) values between *Escherichia* phage ST2, ST4 and closely related phages were below the proposed thresholds for species delineation (Appendix A), indicating that the phage studied belongs to the same *Escherichia*-phage vB-EcoM-PhAPEC2. 

### 3.13. Phylogenetic Analysis 

Phylogenetic analyses between the genomes of related phages were performed with MEGA using the Neighbor-Joining Algorithm. Initially, the first phylogenetic tree was constructed based on the *terL* genes encoding large terminase (TerL) protein with enzymatic activities necessary for phage packaging. One *terL* gene sequence was extracted from *Escherichia* phage ST2 or ST4 genome, while other sequences representing the most similar hits were obtained from the NCBI GenBank database. The second phylogenetic tree involved 10 whole genome sequences of which one sequence of *Escherichia* phage ST2 or ST4, while 9 genomes were obtained from the NCBI GenBank database. 

Both phylogenetic trees confirmed the high similarity of *Escherichia* phage ST2 to *Escherichia* phage vB-EcoM-PhAPEC2 and its close relation to phages of the subfamily *Tevenvirinae*. However, the phylogenetic tree based on the *terL* gene formed distinct clusters and grouped the *Escherichia* phage ST2 with *Escherichia* phage vB EcoM PhAPEC2, WFL6982, WFK, phiC120 and Enterobacteria phage RB69 (Figure 10A), the phylogenetic tree based on the whole genome sequences grouped *Escherichia* phage ST2 with *Escherichia* phage vB EcoM PhAPEC2, WFL6982, WFK, *Escherichia* phage SF, Enterobacteria phage RB69 and *Shigella* phage Shf125875 (Figure 10B). The *Shigella* phage phi25-307, JK42 and *Escherichia* phage vB EcoM JS09 formed distinct cluster (Figure 10B). The BLAST ring image generator (BRIG) (Figure 10C) represents the differences between the 10 phage genomes as low similarity or missing genes are indicated with white gaps. 

Both phylogenetic trees (based on the *terL* genes or whole genome sequences) confirmed the high similarity of *Escherichia* phage ST4 to *Escherichia* phage PGN829.1 and phiG17 and its close relation to phages of the family *Podoviridae* (Figure 11A,B), although, the phylogenetic tree based on the *terL* gene formed distinct clusters and grouped the *Escherichia* phage ST4 with (vB EcoP G7C) (Figure 11A). The BLAST ring image generator (BRIG) (Figure 11C) represents the differences between the 10 phage genomes as low similarity or missing genes are indicated with white gaps.

The comparison between *Escherichia* phage ST2 or ST4 genome and the most similar phages was performed using Mauve software to highlight the differences between them. Due to the high similarity between *Escherichia* phage ST2 and *Escherichia* phage vB EcoM PhAPEC2 or between *Escherichia* phage ST4 and PhAPEC7 and PGN829.1 (based on identity percentage and DDH value), most proteins were homologues with each other, while some blank regions could be detected through genome comparison of these phages (Figure 12 and Figure 13). Additionally, genes which are present only in *Escherichia* phage ST2 (Figure 12 and Table 6) or ST4 (Figure 13 and Table 7) compared to PhAPEC2 or PGN829.1 phages, respectively, were reported. Fifteen genes were detected in *Escherichia* phage ST2 with no similarity to any gene of *Escherichia* phage vB-EcoM-PhAPEC2, in which six were identified as hypothetical proteins (HP) in closely related phages while one was detected in phages belongs to the family *Podoviridae.* Eleven genes were only detected in *Escherichia* phage ST4 compared to PGN829 phage, in which eight genes were identified as hypothetical proteins.

## 4. Discussion

Shiga toxin-producing *E. coli* (STEC), is a food-borne pathogen associated with outbreaks worldwide. It is also considered to be the most substantial etiologic agent of diarrhea causing high morbidity and mortality, specifically in children. Different STEC serotypes are related to human disease, of which serotype O157:H7 which is considered to be the major source of *E. coli* food poisoning outbreaks. Generally, STEC strains are characterized by the presence of main virulence genes which are Shiga toxin- and intimin-encoding genes. The intimin is a 90 kDa protein which contributes to production of the attaching and effacing lesions on intestinal epithelial cells causing acute disease. The *stx* genes encode a family of toxins, including *stx1* and *stx2*. The *stx2* comprises seven recognized variants from *stx2*a to *stx2*g [32]. These toxins are influential cytotoxins causing tissue damage in humans and animals.

Some studies confirmed the presence of enterohaemorrhagic *E. coli* in Egypt. In 2012, El-Leithy et al. [33] isolated STEC O157 from water, and reported that the most prevalent virulence genes in it were *stx2* and *stx1* genes by 98 and 84%, respectively. Selim et al. (2013) [34] also isolated STEC serotypes O157, O158, O125, O114 and O26 from meat products, fresh water, and the feces of people and animals with diarrhea. In 2017, Hamed et al. [35] isolated STEC by 23.68% from minced meat, beef burger, sausage and karish cheese. Abd El-Gany et al. (2020) [36] isolated STEC in a percentage of 44.9% from stool samples of infants suffering from diarrhea, and 35% of these isolates were STEC O157. Finally, Elmonir et al. (2021) [37] isolated STEC from 6, 7, 12 and 10% of collected milk, beef, diarrhetic cattle and human samples, respectively. The O26 and O111 were the most predominant STEC with *stx* genes. 

Lytic bacteriophages are viruses that cause lysis of specific bacterial hosts. Therefore, theses phages have been recommended as novel inoffensive and effective biocontrol agents against bacterial pathogens in different medical and agricultural sectors. 

Prior to lytic phage isolation, it is very imperative to test the bacterial host for lysogeny state through prophage induction test because the presence of prophage provides resistance to the infected bacteria against superinfection by other related phages. Prophages encode proteins called superinfection exclusion proteins which prevent further phage infection through blocking phage binding to the bacterial cell or blocking phage genome injection [38]. *Escherichia coli* O 157:H7 wild type strain 93,111, used as a representative Shiga toxin-producing *E. coli* to isolate STEC phages, was not lysogeny. This was confirmed applying the prophage induction test through irradiation of the cells with UV light. The non-lysogenic bacteria did not exhibit plaques on the bacterial lawn after irradiation. It is known that the stress of prophages using UV radiation can release the phage DNA from the bacterial genome, resulting in phage replication and bacterial cell lysis [38].

In the current study, all phages active against STEC O157:H7 wild type strain 93,111 were isolated from sewage. Generally, sewage contains a large diversity of coliforms due to the fecal contamination, and therefore it is considered to be a natural reservoir of enteric pathogens and their phages [39]. This supports our results which confirmed that the sewage is a superior representative source to isolate bacteriophages infecting *E. coli* O157:H7. 

Generally, the phage classification depends principally on both morphological and genetic characterization. Morphologically, according to the guidelines of the International Committee on Taxonomy of Viruses, it could be concluded that the STEC P1 and P3 may be classified as members belonging to the *Tectiviridae* or *Corticoviridae* families. 

Alternatively, the STEC P2 could be considered to belong the family *Myoviridae* or *Siphoviridae.* The genomic analysis confirmed its belonging to the subfamily *Tevenvirinae* under family *Myoviridae*. The morphology of STEC P4 suggested that it could be a member in the family *Podoviridae*. The genomic characterization was in consent to the results of TEM examination, and further classified the phage under the subfamily *Gamaleyavirus*. Rodríguez-Rubio et al. in 2020 reported that most STEC phages are classified into two families of the order *Caudovirales*, which are *Podoviridae* and *Siphoviridae* [40], while other studies described some STEC phages as members of *Myoviridae* [22,41,42]. Although some studies [43,44,45] reported that all known Stx phages have a common head-tail structure ranging from icosahedral or elongated head with contractile or non-contractile tail with or without tail fibers, two tailless phages (STEC P1 and P3) were isolated in this study. 

The host range is one of the most considerable biological characteristics of bacteriophages. It is very important to determine the host range of phages to assess their applicability as antibacterial agents in phage therapy, food bio-preservation and in protection of industrial fermenters from bacterial contamination. The results exhibited that all tested toxigenic *E. coli* O157 (STEC O157:H7 ATCC 35150 and O157 86.24) and nontoxigenic *E. coli* O157 ATCC 100700 were sensitive to all isolated phages, suggesting that these phages have a relatively high specificity to *E. coli* O157. The specificity of STEC phages to *E. coli* O157:H7 was reported by other studies [22,41,42]. The STEC P2 has a comparatively broad host range as it has lytic activity not only against different toxigenic and nontoxigenic *E. coli* strains but also against *Shigella boydii*, signifying that it can infect members of other genera closely related to its host. *Shigella boydii* is one of the causative agents of dysentery in humans worldwide, especially in developing countries through fecal–oral contamination. Consequently, STEC P2 could be considered as a natural food preservative used to control specific food-borne pathogens including different STEC strains and *S. boydii* simultaneously. 

The ability of STEC P2 to infect both STEC and *S. boydii* indicates the presence of different host specificity associated genes in the phage genome which encode tail proteins. The tail proteins are known as proteins for determination of host specificity. Four ORFs in the STEC P2 genome produce tail proteins including tail fibers, straight tail fiber and long tail fibers. 

The lytic capability and host range are considered as key features for the evaluation and selection of phages for controlling bacterial pathogens. There are different methods to evaluate the phage lytic activity from which PhageScore, streak-based method and Lysis Score which was applied in this study with simple modification [46,47,48]. In the Lysis Score method, description of phage lysis activity depends on OD changes with time. The phage activity has a lysis score ranging between 1–3. Score 1 is specified for lysis of bacterial culture at the highest used MOI, score 2 for lysis at two higher MOIs, and 3 for lysis at three MOIs [46]. The changes in log cfu counts were applied instead of OD changes to assess the lytic activity of STEC P2 and P4. According to the results, the STEC P4 received a specified lysis score of 2 as the complete bacterial lysis was recorded at the two highest MOIs of 0.1 and 1 pfu/cfu. The results also revealed that STEC P2 could be considered more efficient in killing its host as the complete bacterial lysis appeared after 6 h with MOI of 0.1. With the same MOI of STEC P4, the complete lysis appeared after 7 h. 

Using phages in the biocontrol of pathogens in food is becoming progressively accepted as a green technology targeting bacterial pathogens specifically without changing food properties or inducing harmful effects in human health. The application of bacteriophages as food bio-preservatives necessitates evaluating the stability of their lytic activity in different environmental conditions. These conditions, such as radiation, temperature, acidity, alkalinity and salinity, can affect the viability and infectivity of phages as they can damage phage structure and nucleic acid. Naturally, phages in food will be subjected to ingestion, so it is imperative to evaluate phage viability in the gastrointestinal tract. In this study, the phage survivability in relation to environmental factors of UV radiation, high temperature and pH was appraised. The phage activity was also estimated in bile salt, simulated gastric and intestinal fluids. 

The results demonstrated that the long-time exposure to UV radiation decreases the phage lytic activity reaching 89.11% reduction in STEC P2 lysate after 75 min. Ramirez et al. (2018) [8] reported that the cocktail of *E. coli* O157:H7 phages; FJLA23, FKP26, FC119 and FE142 lost its lytic activity after 15 min. 

High temperatures play a critical role in the bacteriophage survival, capacity for attachment and length of the latent period [49]. The results indicated that the STEC P2 was considerably sensitive to heating as the significant decrease in phage count was observed after all heat treatments. The tested phages lost their infectivity entirely after boiling, indicating the temperature of 100 °C has a lethal effect against isolated STEC P2 and P4. On the other hand, the phages retained higher lytic competence of 90.4 and 92.68% for STEC P2 and P4, respectively, with the HTST pasteurization comparing with the LTLT pasteurization signifying that the prolonged exposure to high temperature is relatively more effective than short exposure to higher temperature. This result is consistent with previous studies [8,50,51]. On the other hand, Ullah et al. (2021) [52] reported that the coliphage AS1 displayed optimum activity at high temperatures of 50 °C and 60 °C/30 min, and its infectivity reduced at 70 °C for the same exposure time. In 2007, Caldeira and Peabody [53] proposed that the disulfide bonds between capsid proteins may have a role to protect phages from thermal denaturation. 

The bacteriophages may persist in acidic or alkaline conditions. Wunsche et al. (1989) [54] reported that the pH has an effect on the adsorption rate of phages on the host cell through changing the charge of capsid protein. Ullah et al. (2021) [52] stated that most phages remain viable in a pH ranging from 5–9. In this study, there is no single pattern to evaluate the survivability of the two phages in different pH values. Comparatively, the STEC P2 can retain significant higher lytic activity of 97.42 and 95.67% in alkaline conditions of pH 9 and 11, respectively, rather than STEC P4, which displayed high activity of 84.73 and 85.96% in acidic conditions of pH 3 and 5, correspondingly. Similar results were observed in the study of Coffey et al. (2011) [55], as they found that *E. coli* O157:H7 phage e11/2 was more stable under acidic conditions (pH 3), whereas the e4/1c phage was stable in alkaline conditions (pH 11 and 12).

In the present study, it was also recorded that the pH 1 and 13 had a lethal effect against STEC P2 and P4, respectively. This may be attributed to the denaturation of phage proteins in strong acidic or alkaline conditions. This observation confirmed presence of differences in the structural proteins between the two phages.

To a large extent, STEC P2 and P4 were resistant to gastric conditions, represented by acidic pH and presence of pepsin enzyme, as their survivability reduced only to 80.17 and 73.41%, respectively, after 4 h. In intestinal conditions (neutral pH of 6.8 and presence of trypsin enzyme), the survivability reduced only to 87.53% for STEC P2 and to 87.45% for STEC P4 after 4 h. Generally, the reduction of phage survival in gastric fluid may be attributed to the denaturation of phage proteins caused by extreme acidity. The presence of proteolytic enzymes in gastric and intestinal fluids may also contribute to the denaturation of phage proteins [8].

It is important to analyze the complete genome sequence of phages to understand their biology and to evaluate their potential application in food preservation. 

The annotation of *Escherichia* phage ST2 genome revealed that 229 CDS (84.81%) are expressed to produce functional proteins categorized into functional groups of DNA replication, transcription regulation, phage structure and host lysis. The DNA metabolism/replication/modification proteins include deoxynucleoside monophosphate kinase, 3′-phosphatase, 5′-polynucleotide kinase, thymidine kinase, DNA topoisomerase, DNA polymerase, sliding clamp DNA polymerase accessory protein, replication factor C small subunit/phage DNA, DNA helicase, DNA primase, DNA primase/helicase, endonuclease, recombination related endonucleases, exonuclease, ribonucleotide reductase of class III and Ia, DNA end protector protein and single stranded DNA binding protein. 

The transcription regulation proteins comprise endoribonuclease translational repressor, RNA polymerase binding protein, RNA polymerase sigma factor, transcriptional regulator, transcription regulator MotA, ModA or ModB ribosyltrasferase and RNA polymerase-ADP-ribosyltransferase. After entering the host bacterial cell, the phage produces proteins to interact with the key enzymes of the bacteria to inhibit or modify related biological activities. RNA polymerase is the predominant target to use bacterial transcription machinery in the initial stage of attack and to inhibit the host RNA polymerase activity in the subsequent stages [56].

The phage genome encodes lysozyme and DNA ejectosome component gp16 which is considered to facilitate phage DNA injection into the host cell. During infection, numerous internal capsid proteins (gp14, gp15, and gp16) are ejected into the host cell wall and form an ejectosome, allowing the translocation of phage DNA from the capsid into the bacterial cytoplasm [57].

The proteins for phage head morphogenesis, tail component formation and virion assembly involve prohead core protein, prohead assembly proteins, major capsid protein, capsid vertex protein, capsid and scaffold proteins, neck proteins, tail tube, tail sheath proteins, tail fiber proteins, long tail fibers, tail connector protein, baseplate hub structural proteins, baseplate wedge initiator, baseplate wedge tail fiber connector, baseplate wedge subunit, baseplate wedge subunit and tail pins, baseplate tail tube cap, phage straight tail fiber, head completion protein, tail completion protein, portal vertex protein, head assembly chaperone protein, tail assembly protein and fibritin. Fibritin is known as a structural protein for specific phage-assembly process. After the phage head is connected to the tail, six fibritin molecules attach to the neck to form a collar with six fibers. It also promotes the assembly and attachment of long tail fibers to the tail baseplate. Fibritin is considered as a sensing device to control the shrinkage of long tail fibers in adverse environments and prevents infection [58].

The proteins involved in host cell lysis are holin, holin class II, endolysin and outer membrane lipoprotein Rz1. Generally, in bacteriophages, the host lysis depends on holin–endolysin lysis system. In T-even phages, specifically T4 phages, the holins (T) have cytoplasmic, periplasmic domains and single transmembrane domain. At a specific genetically programmed time of the infection cycle, holins cause a lethal process called holin triggering in which the polymerization of transmembrane domains forms stable and non-specific micron-scale holes across the cell membrane resulting in cell membrane permeabilization and termination of respiration and macromolecular synthesis [59]. These holes allow the endolysin to gain access to the cell wall and degrade peptidoglycan. In phages of G^-^ bacteria, the cell membrane, peptidoglycan and outer membrane must be disrupted for complete cell lysis which is achieved by the third functional lysis proteins called spanins [60]. In *Escherichia* phage ST2, the outer membrane lipoprotein Rz1 (o-spanin) was found for lipoprotein lysis. Phage phospholipase was determined suggesting that it maybe have a role for lysis of phospholipid of the outer membrane. 

Furthermore, it was found that the phage genome encodes lysis inhibition (LIN) proteins of rI and rIIA lysis inhibition regulators. The lysis inhibition, which is considered to be a unique attribute of the T-even family phages, is defined as the ability of phages to delay lysis for several hours in response to superinfection of bacterial cell with other related phages. This superinfection activates phage antiholin protein (rI lysis inhibition regulator), which plays a key role in lysis inhibition, to bind the periplasmic domain of T holin and form heterotetrameric 2RI-2T complex preventing holin triggering and therefore extending the infection cycle and accumulating the new virions into the host cell [61]. Some studies [62,63,64] confirmed that the rIIA proteins is not directly involved in lysis inhibition. They seem to play a role in inhibition of all energy-dependent processes including gene expression, Mg^2+^ transport and ATP biosynthesis, or they are required for maintaining membrane integrity.

Finally, there are some proteins with additional functions as dCMP deaminase, thioredoxin and glutaredoxin.

Comparing to *Escherichia* phage ST2, the annotation of *Escherichia* phage ST4 genome displayed that, only 30 CDS (35.29%) were predicted to produce different functional proteins. These proteins represent only DNA polymerase, helicase and primase for DNA replication, and RNA polymerase for transcription regulation. 

Interestingly, this genome has genes coding for superinfection exclusion protein, integrase and antirepressor protein which are considered as markers of temperate phages. Generally, the lysogenic bacterial cell is resistant to subsequent infection by other phages closely related to the integrated prophage by superinfection exclusion (Sie) proteins (membrane-associated proteins) encoded by phage or prophage. The Sie proteins prevent further phage infection through different mechanisms including blocking phage binding to the host cell, blocking phage genome injection or preventing degradation of the cell wall [65]. The antirepressor protein has a role for conversion of prophages from lysogenic to lytic mode. The phage repressor which prevents expression of lytic genes in the prophage could be inactivated by antirepressors allowing prophages to switch from a lysogenic to a lytic cycle [66]. Usually, the presence of these proteins suggests the weak ability of *Escherichia* phage ST4 for lysis of its host. This finding is supported by the appearance of small plaques of ST4 on the bacterial lawn comparing with ST2. Moreover, the presence of these proteins could indicate that *Escherichia* phage ST4 may be converted accidentally from lysogenic phage through prophage induction under stress environmental conditions. 

Significantly, the genome analysis confirmed that both studied phages are free of genes associated with toxins or other virulence factors. 

Finally, the results of in vitro studies and genome analysis showed the stability of the two phages, with different degrees, in the hostile conditions represented by UV irradiation, high temperatures, pH, bile salt, gastric and intestinal fluids supporting their potential application as food bio-preservatives against Shiga toxin-producing *E. coli*. In vivo studies to control STEC using the tested phages are recommended for complete characterization to be used in food preservation. 

## Figures and Tables

**Figure 1 biology-11-01180-f001:**
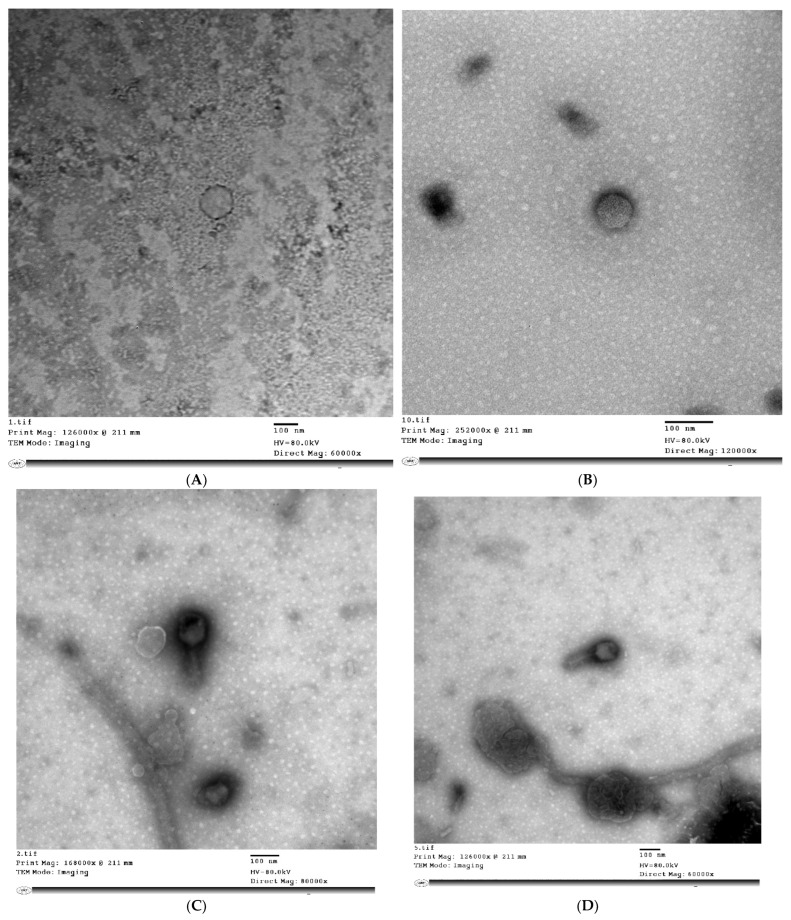
Transmission electron microscopy (TEM) micrographs of *E. coli* phage particles (STEC P1–STEC P4) stained negatively by 2% (*w*/*v*) phosphotungstate at pH 7.2. STEC P1: (**A**) direct magnification: 60,000×; STEC P1: (**B**) direct magnification: 120,000×; STEC P2: (**C**) direct magnification: 80,000×; STEC P2: (**D**) direct magnification: 60,000×; STEC P3: (**E**) direct magnification: 10,000×; STEC P3: (**F**) direct magnification: 80,000×; STEC P4: (**G**) direct magnification: 80,000×.

**Figure 2 biology-11-01180-f002:**
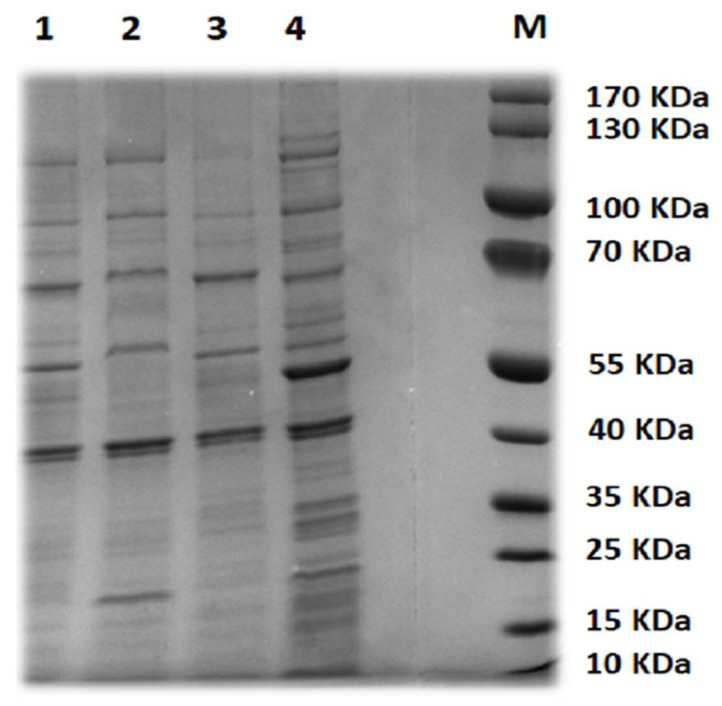
SDS-PAGE analysis of phage structural proteins. Lane M, protein marker. Lanes 1–4, structure proteins from STEC P1–STEC P2.

**Figure 3 biology-11-01180-f003:**
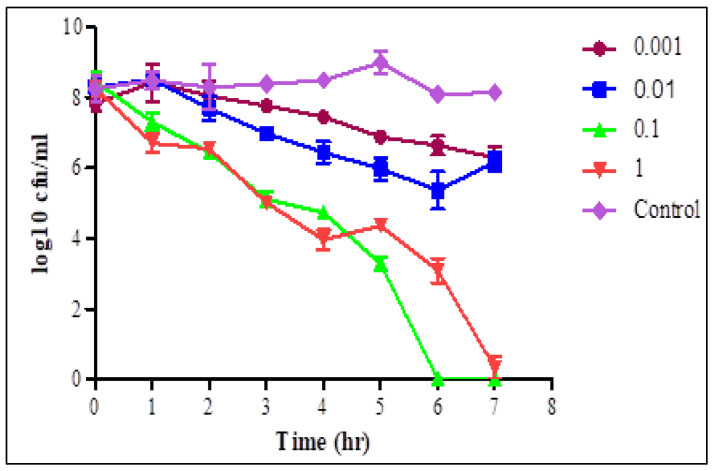
Lytic activity of STEC P2 after infection of STEC O157:H7 wild type strain 93,111 at MOI of 1, 0.1, 0.01 and 0.001 for 7 h. LSD value at 0.05: 0.7.

**Figure 4 biology-11-01180-f004:**
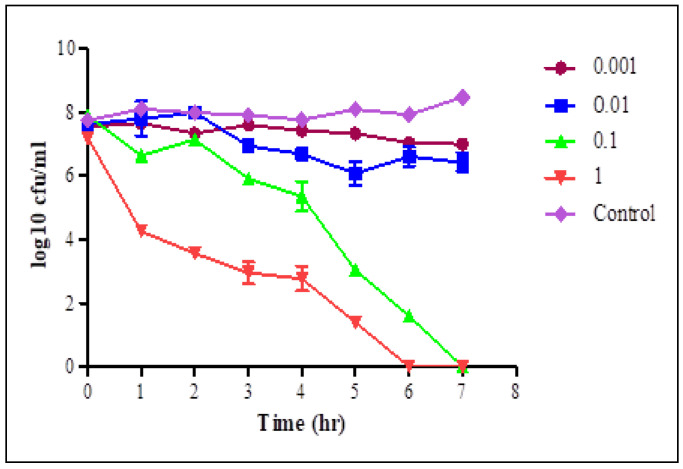
Lytic activity of STEC P4 after infection of STEC O157:H7 wild type strain 93,111 at MOI of 1, 0.1, 0.01 and 0.001 for 7 h. LSD value at 0.05: 0.523.

**Figure 5 biology-11-01180-f005:**
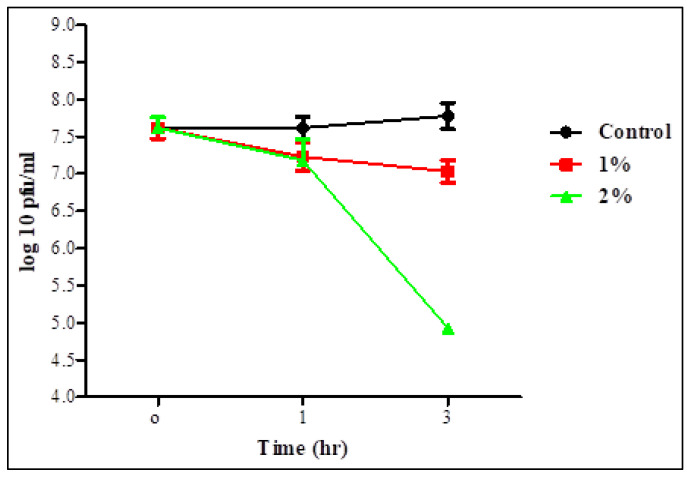
STEC P2 stability in 1 and 2% bile salt. LSD value at 0.05:0.296.

**Figure 6 biology-11-01180-f006:**
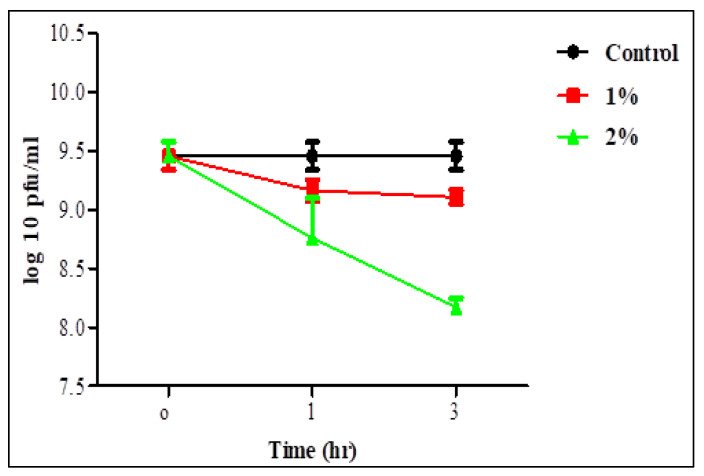
STEC P4 stability in 1 and 2% bile salt. LSD value at 0.05:0.296.

**Figure 7 biology-11-01180-f007:**
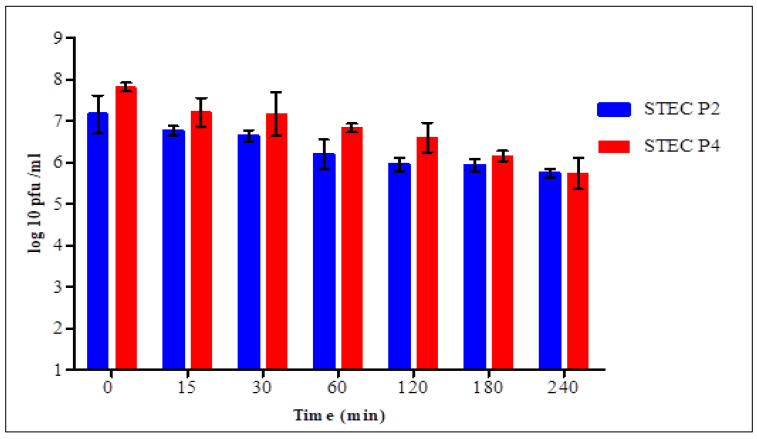
Effect of simulated gastric fluid on the counts of STEC P2 and STEC P4. LSD value at 0.05:0.286.

**Figure 8 biology-11-01180-f008:**
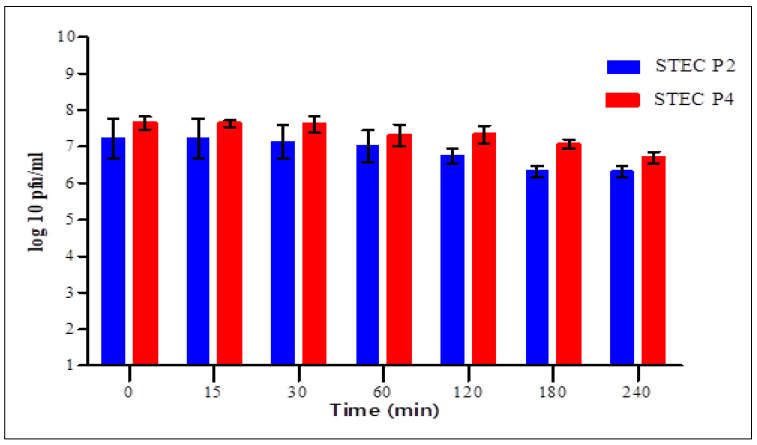
Effect of simulated intestinal fluid on the counts of STEC P2 and STEC P4. LSD value at 0.05:0.449.

**Figure 9 biology-11-01180-f009:**
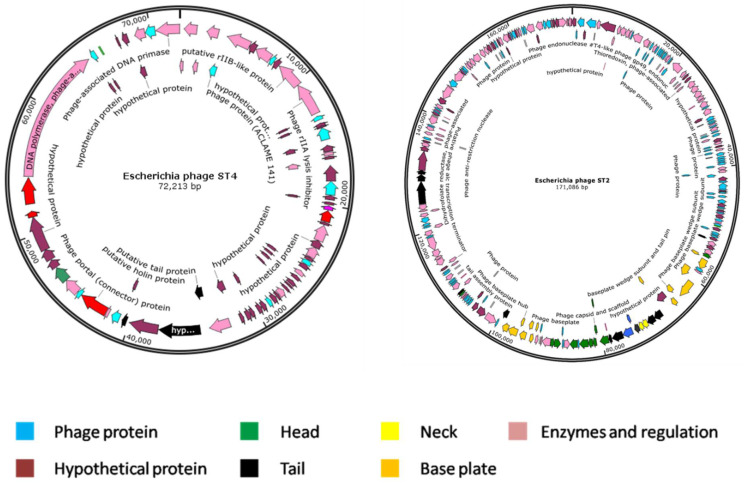
Map of the genome organization of *Escherichia* phage ST2 and ST4. Different structural proteins are indicated with different colors.

**Figure 10 biology-11-01180-f010:**
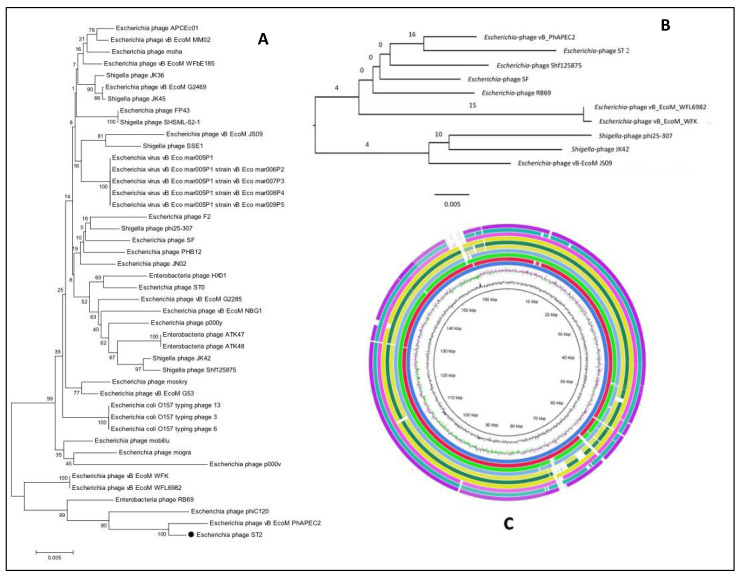
(**A**) phylogenetic tree of *Escherichia* phage ST2 based on the sequence of terL gene, (**B**) shows the phylogenetic GBDP tree and (**C**) genome comparisons with the closest phages obtained from the NCBI GenBank using BRIG software version 0.95.

**Figure 11 biology-11-01180-f011:**
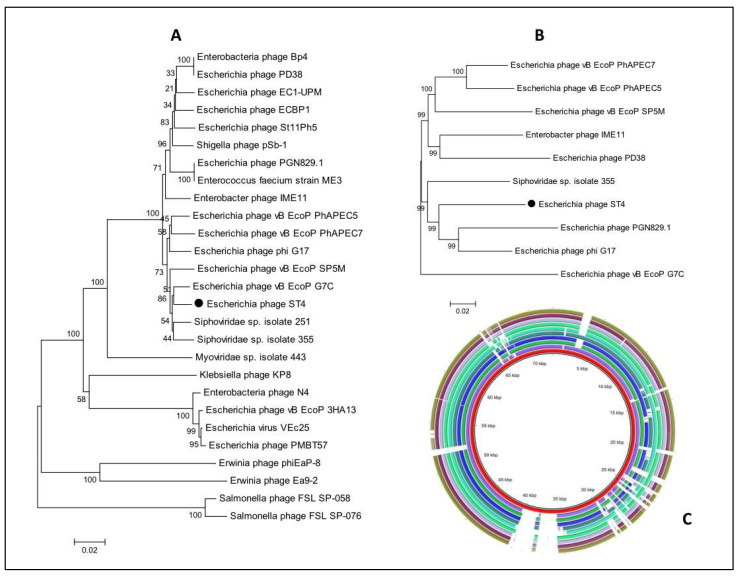
(**A**) phylogenetic tree of *Escherichia* phage ST4 based on the sequence of *terL* gene, (**B**) shows the phylogenetic GBDP tree and (**C**) genome comparisons with the closest phages obtained from the NCBI GenBank using BRIG software version 0.95.

**Figure 12 biology-11-01180-f012:**
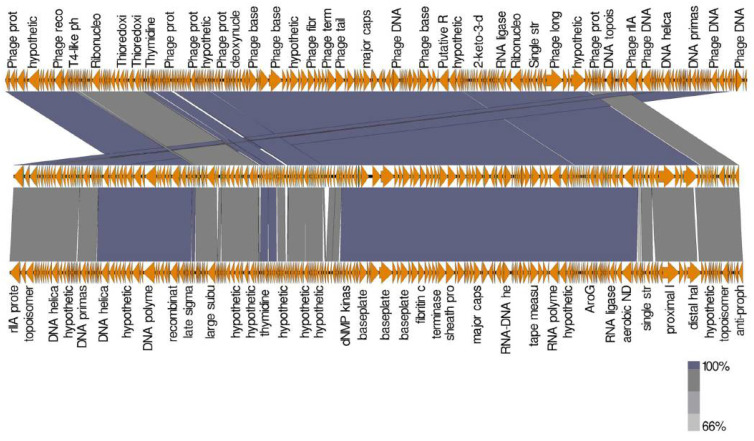
Comparison of ORFs between *Escherichia* phage ST2 (**above**) and the most closed phages *Escherichia* phage vB EcoM PhAPEC2 (**middle**) and *Shigella* phage Shf125875 (**below**). Analysis was performed using EasyFig software version 2.2.5.

**Figure 13 biology-11-01180-f013:**
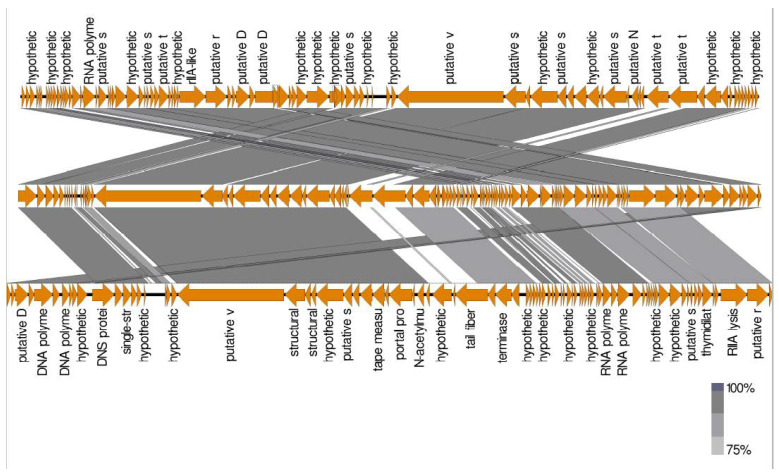
Comparison of ORFs between *Escherichia* phage ST4 (**middle**) and the most closed phages *Escherichia* phage vB EcoP PhAPEC7 (**above**) and *Escherichia* phage PGN829.1 (**below**). Analysis was performed using EasyFig software version 2.2.5.

**Table 1 biology-11-01180-t001:** Host range of selected STEC phages.

Bacterial Strains	Isolated Coliphages			
	STEC P1	STEC P2	STEC P3	STEC P4
Shiga toxin-producing *Escherichia coli*				
*Escherichia coli* O157:H7 ATCC 35150	+	+	+	+
*Escherichia coli* O157 86.24	+	+	+	+
*Escherichia coli* O103 87-293	+	+	+	+
*Escherichia coli* O26 Decaf	−	+	−	−
*Escherichia coli* O145 6940	−	−	−	+
*Escherichia coli* O111 TB226A	−	−	−	−
*Escherichia coli* O45 4309-6	−	−	−	−
Nontoxigenic *Escherichia coli*				
*Escherichia coli* O157 ATCC 100700	+	+	+	+
*Escherichia coli* NRRL B-3008	−	+	+	−
*Escherichia coli* K-12 MG1655: b2748	−	+	−	−
*Escherichia coli* B-51077	−	−	−	−
*Escherichia coli* ATCC 35218	−	−	−	−
*Escherichia coli* ATCC 8739	−	−	−	−
Other G^-^ bacteria				
*Salmonella typhimurium* ATCC 14028	−	−	−	−
*Enterobacter aerogenes* B-14144	−	−	−	−
*Shigella sonni* ATCC 29930	−	−	−	−
*Shigella boydii* *	−	+	−	−
*Klebsiella pneumoniae* *	−	−	−	−
*Klebsiella pneumoniae* *	−	−	−	−
*Klebsiella quasipneumoniae* *	−	−	−	−
*Enterobacter cloacae* *	−	−	−	−
*Enterobacter cloacae* *	−	−	−	−
*Enterobacter asburiae* *	−	−	−	−
*Enterobacter ludwigii* *	−	−	−	−
*Enterobacter hormaechei* subsp. *Xiangfangensis* *	−	−	−	−

+, susceptible; −, resistant; *, local identified bacterial strains according to 16 srRNA sequencing and obtained from Agricultural Microbiology Dept., Fac, Agric., Cairo Univ.

**Table 2 biology-11-01180-t002:** MOI of STEC P2 and STEC P4.

MOI (pfu/cfu)	Phage Count (Log Average ± S.D.)	
	STEC P2	STEC P4
1 (group 1)	9.54 ± 0.50 ^cd^	10.67 ± 0.17 ^a^
0.1 (group 2)	10.64 ± 0.15 ^a^	9.94 ± 0.29 ^bc^
0.01 (group 3)	10.44 ± 0.30 ^ab^	9.24 ± 0.49 ^de^
0.001 (group 4)	8.69 ± 0.27 ^e^	6.82 ± 0.37 ^f^

Statistically significant differences are indicated by different superscript letters.

**Table 3 biology-11-01180-t003:** UV stability of STEC P2 and STEC P4.

Time (min)	STEC P2		STEC P4	
	Surviving Virions (Log Average ± S.D.)	Bacteriophage Reduction (%)	Surviving Virions (Log Average ± S.D.)	Bacteriophage Reduction (%)
Zero time	10.13 ± 0.24 ^a^	0.0	9.15 ± 0.23 ^b^	0.0
15	6.91 ± 0.76 ^c^	31.68	6.71 ± 0.07 ^c^	26.67
30	5.40 ± 0.66 ^d^	46.54	5.50 ± 0.36 ^d^	39.89
45	3.78 ± 0.1.06 ^ef^	62.57	4.44 ± 0.36 ^e^	51.58
60	1.80 ± 0.1 ^hi^	82.18	3.43 ± 0.68 ^fg^	62.55
75	1.10 ± 0.1 ^i^	89.11	2.61 ± 0.61 ^gh^	71.51

Statistically significant differences are indicated by different superscript letters.

**Table 4 biology-11-01180-t004:** Thermal stability of STEC P2 and STEC P4.

Time	STEC P2		STEC P4	
	Surviving Virions (Log Average ± S.D.)	Bacteriophage Reduction (%)	Surviving Virions (Log Average ± S.D.)	Bacteriophage Reduction (%)
Zero time	11.65 ± 0.34 ^a^	0.0	10.38 ± 0.53 ^b^	0.0
72 °C/15 s	10.53 ± 0.49 ^b^	9.6	9.62 ± 0.15 ^bc^	7.32
63 °C/30 min	5.30 ± 0.73 ^d^	53.1	8.73 ± 0.67 ^c^	18.89
100 °C/10 min	0.00	100	0.00	100
100 °C/20 min	0.00	100	0.00	100
100 °C/30 min	0.00	100	0.00	100

Statistically significant differences are indicated by different superscript letters.

**Table 5 biology-11-01180-t005:** pH stability of STEC P2 and STEC P4.

pH	STEC P2		STEC P4	
	Surviving Virions (Log Average ± S.D.)	Bacteriophage Reduction (%)	Surviving Virions (Log Average ± S.D.)	Bacteriophage Reduction (%)
7	9.70 ± 0.1 ^a^	0.0	9.69 ± 0.17 ^a^	0.0
1	0.00 ^i^	100	7.73 ± 0.34 ^f^	20.23
3	6.60 ± 0.0 ^g^	31.96	8.21 ± 0.34 ^de^	15.27
5	8.80 ± 0.2 ^c^	9.28	8.33 ± 0.064 ^d^	14.04
9	9.45 ± 0.0 ^ab^	2.58	8.03 ± 0.32 ^e^	17.13
11	9.28 ± 0.1 ^b^	4.33	7.95 ± 0.31 ^ef^	17.96
13	1.88 ± 0.12 ^h^	80.62	0.00 ^i^	100

Statistically significant differences are indicated by different superscript letters.

**Table 6 biology-11-01180-t006:** Genes occurred in *Escherichia* phage ST2 and not present in *Escherichia* phage vB EcomM PhAPEC2.

Location on *Escherichia* Phage ST2	Size bp	Closest Hits from NCBI	Query Cover	Similarity%	Function
16,103–16,372	270	*Shigella* phage JK42	100	100	HP
16,992–17,231	240	*Escherichia* phage moskry	100	99.58	HP
33,907–34,113	207	*Shigella* phage JK42	100	100	HP
34,110–34,322	213	*Escherichia* phage vB EcoM KAW3E185	100	99.06	HP
38,256–38,585	330	*Escherichia* phage moskry	100	99.39	HP
38,611–38,925	315	*Escherichia* phage vB-EcoM-NBG1	100	99.37	HP
47,138–47,302	165	*Shigella* phage JK42	100	99.39	HP
167,349–167,552	204	*Klebsiella* phage Patroon	100	89.71	Holin
167,590–167,976	387	No significant similarity	-	-	Single-stranded DNA-binding protein
167,998–168,645	648	No significant similarity	-	-	Tail fiber protein
168,675–170,066	1392	*Yersinia* phage phiYe-F10 (*Caudovirales)*	99	97.77	Internal virion protein D
170,080–170,235	156	*Citrobacter* phage SH2*(Caudovirales)*	100	100	HP
170,336–170,437	102	*Serratia* phage SM9-3Y (*Podoviridae)*	100	100	HP
170,448–170,903	456		100	99.34	N-acetylmuramoyl-L-alanine amidase
170,982–171,086	105		85	96.67	No significant similarity found

HP, hypothetical protein.

**Table 7 biology-11-01180-t007:** Genes occurred in *Escherichia* phage ST4 and not present in *Escherichia* phage PGN829.

Location on *Escherichia* Phage ST4	Size bp	Closest Hits from NCBI	Query Cover	Similarity%	Function
2–331	330	*Escherichia* phage vB EcoP G7C	100	98.48	Phage-associated DNA primase
24,211–24,444	234	No significant similarity	-	-	HP
24,472–24,822	351	*Siphoviridae* sp.	98	95.36	Phage protein
25,203–25,445	243	*Escherichia* phage vB EcoP G7C	100	96.3	HP
25,753–25,935	183	No significant similarity	-	-	HP
25,932–26,174	243	No significant similarity	-	-	HP
27,946–28,152	207	*Enterobacter* phage IME11	33	98.57	HP
29,561–29,902	342	*Siphoviridae* sp. isolate 355	91	95.19	HP
34,547–37,714	3168	No significant similarity	-	-	HP
37,768–39,996	2229	*Escherichia albertii* strain 2012EL-1823B	60	85.41	Peptidase
40,002–40,151	150	No significant similarity	-	-	HP

HP, hypothetical protein.

## Data Availability

The genome sequence associated with this paper is available in NCBI GeneBank with the name of *Escherichia* phage ST2 (accession number: OM982647) and *Escherichia* phage ST4 (accession number: OM982646).

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
