# Peer review of "Evaluating the Phenotypic and Genomic Characterization of Some Egyptian Phages Infecting Shiga Toxin-Producing Escherichia coli O157:H7 for the Prospective Application in Food Bio-Preservation"

_biology, 2022, doi:10.3390/biology11081180_

Round 1
Reviewer 1 Report
There are some odd highlights in the document that need to be addressed (see line 57 and the word Shiga throughout).
Figure 2 could use some improvement. The top axis is difficult to read. The fonts along various axis are squashed and distorted. Please correct.
Figure 3. Same issue -distorted figures and low resolution. The resolution and distortion issue impacts all figures.
Figures 10 and 11 could probably be paired down to better make your point.
Figure 12 seems pointless without some conclusion drawn from it.
Table 6 caption is above the table. This is rampant and confusing. Fix. Check caption syntax and language for necessary edits,
Reviewer 2 Report
All of the comments I made were addressed in this draft.
Reviewer 3 Report
The authors isolated and characterized two phages that infect Shiga Toxin-Producing E.coli (STEC) O157:H7, one of the major food-borne pathogens. With the comprehensive assessment of the properties of these two phages, the authors propose that these two phages could be applied in food biopreservation.
This is a well-documented manuscript, covering a wide range of characterizations of the two new STEC-infecting phages. However, the phages presented here seem to be not specific to (STEC) O157:H7, but also to other Shiga toxin-producing and also non-pathological E.coli, which makes them more board and generic E.coli-targeting phages. The claim of prospective application of these phages comes from their insusceptibility to harsh conditions, including pH and bile salt, and simulated gastric and intestinal fluid. However, a more thorough analysis of animal studies should be included to truly demonstrate their applicability to food preservation. But the data presented here is indeed the first step toward the goal. I have minor points for authors to address before considering publication.
-
In Figure1, the information for each TEM image should be clear and legible.
-
The statistic analysis of the data throughout the paper is not clear to me, and in the method session, please specify which program the authors used. The “statistically significant differences are indicated by different letters” throughout the paper, could the author specify what each letter means?
-
Figure 7 is more squeezed than Figure 8, please correct this.
-
The paper is well written but very lengthy. I suggest the authors at least shorten the discussion part and focus on more the potential pitfalls of the application of the phages presented here and the future directions of phages in the food-preservation industry.
Round 2
Reviewer 1 Report
Done already. Improvements are fine.
This manuscript is a resubmission of an earlier submission. The following is a list of the peer review reports and author responses from that submission.
Round 1
Reviewer 1 Report
In the submitted manuscript, El-Sayed et. al. investigated using phages as a tool against SETC. The authors isolated four anti-SETC phages from environment (i.e., sewage and tap water) and analyzed their anti-SETC activity and host-range of two phages. In addition, authors examined the survival of these phages in difference conditions that related to therapeutic application. Overall, the study is well conducted and will provide useful information to the readers in the field.
Some minors suggestions for authors to consider:
Throughout the manuscript, there are many short-sentence paragraphs, for instance, line 429 - 431, line 713- 714, etc. I suggest the authors combine these sentences/short paragraph into adjacent paragraph.
Reviewer 2 Report
The paper needs to be rewritten to address passive voice and sentence structure for English audiences. It is rampant through out the Simple Summary and Abstract. The simple summary and abstract are the same length. The abstract is easier to understand than the simple summary.
"The most important STEC serotypes associated with food 50
poisoning outbreaks in the world are O157, O146, O145, O128, O121, O118, O117, O113, 51
O111, O103, O91 and O26 [3]."
While serogroups are useful for understanding potential risk in some cases, it is increasingly recognized that virulence markers are better ways of determining virulence potential than are o-groups.
https://www.ncbi.nlm.nih.gov/pmc/articles/PMC2238209/
https://www.nature.com/articles/nrmicro818
We should embrace this idea while recognizing the utility of o-groups sans better sources historically.
I am impressed with the use of the word "thence" in line 56. However, it's archaic. Write with simpler words that are direct and concise.
Line 136 states "This strain" However, I can cast my gaze upward for some time before learning which strain "this strain" is. Name the strain.
Methods!
"Different water samples were screened for the presence of phages infecting shiga- 143
toxin producing E. coli O157:H7. Fifteen water samples comprised 11 sewage, and 4 tap 144
water samples were collected from Giza Governorate using sterile 500 mL bottles. Prior 145
to phage isolation, the sewage samples were filtered through filter paper to remove the 146
impurities."
-What impurities are being removed? Why is this important?
"The host range of all purified phages was determined applying the spot test against 197 25 G" against what? 25 what? What is a G? Its a gravitational constant.
Fix all figure caption formatting. They're justified right. Captions are not sufficiently explanatory and are confusing.
Captions should be below tables and figures.
There's a lot of good work here, but the body does not support your conclusions or hypothesis sufficiently.
Reviewer 3 Report
This study described four STEC O157:H7 lytic phages isolated from sewage. Two phages were further analyzed by biologic and genomic characterizing.
In this study, there is no evaluation on controlling STEC in food, so the title ‘… as Biocontrol Agents in Food’ and the purpose ‘evaluating the applicability of phages, as prospective food bio-preservatives’ were not quite fit.
Major concerns:
- The stx1/stx2 genes are generally carried by prophages (usually called Stx phages). In this study, STEC O157:H7 wild type strain 93111, encoding stx1, stx2 and eae genes, was used for phage isolation, and the non-lysogenic STEC O157:H7 were obtained through UV irradiation. So, how about the stx1/stx2 (Stx phages) in wild 93111 (and after irradiation)?
- In figure S1, Did NTC mean?
Other comments:
- Shiga toxin, always capitalize ‘S’ throughout the manuscript.
- Figure 10, no bootstrap values (A) or low bootstrap values (B).
- Figure 11A, Escherichia phage STEC-ED4? (ST4 or P4?).
Reviewer 4 Report
The manuscript describes the isolation and characterization of bacteriophages targeting STEC E. coli strains for the potential use in food safety applications. The paper is straight forward and the conclusions sound. All the parameters tested are needed to determine if the phage would be stable in food storage situations. Furthermore, the genome sequence is necessary to show the phages do not carry toxin genes. The methods need some additional detail, and I have highlighted specific comments below.
Line 120: Please clarify – was this SYBR Green PCR?
Line 136-140: Were the cells in buffer or in medium when irradiated? How many cells were in irradiated suspension?
Lines 143-147: What volumes of water were collected?
Line 162: Give the concentration and pH of the phosphate buffer
Line 198 and elsewhere: Please write out Gram-negative
Lines 223 – 225: It says 5 ml of lysate. Was this in a tube, open plate, …?
Lines 239 – 241: What concentration of bile salts? Was this in a buffer or in water?
Line 359: What are these groups? Are they defined in the text?
Table 2: Define the superscript numbers.
Lines 382 – 385: These 2 sentences conflict with each other.